# Fabrication, Structure and Functional Characterizations of pH-Responsive Hydrogels Derived from Phytoglycogen

**DOI:** 10.3390/foods10112653

**Published:** 2021-11-01

**Authors:** Xiuting Hu, Yao Liu, Yimei Chen, Tao Zhang, Ming Miao

**Affiliations:** 1State Key Laboratory of Food Science and Technology, Jiangnan University, Wuxi 214122, China; xthu@ncu.edu.cn (X.H.); 6170111031@stu.jiangnan.edu.cn (Y.L.); 6190112144@stu.jiangnan.edu.cn (Y.C.); zhangtao@jiangnan.edu.cn (T.Z.); 2State Key Laboratory of Food Science and Technology, Nanchang University, Nanchang 330047, China

**Keywords:** phytoglycogen, pH-responsive hydrogel, chain elongation, structure, characterization

## Abstract

The pH-responsive hydrogels were obtained through successive carboxymethylation and phosphorylase elongatation of phytoglycogen and their structure and functional characterizations were investigated. Phytoglycogen (PG) was first carboxymethylated to obtain carboxymethyl phytoglycogen (CM-PG) with degree of substitution (DS) at 0.15, 0.25, 0.30, and 0.40, respectively. Iodine staining and X-ray diffraction analysis suggested that the linear glucan chains were successfully phosphorylase-elongated from the non-reducing ends at the CM-PG surface and assembled into the double helical segments, leading to formation of the hydrogel. The DS of CM-PG significantly influenced elongation of glucan chains. Specifically, fewer glucan chains were elongated for CM-PG with higher DS and the final glucan chains were shorter, resulting in lower gelation rate of chain-elongated CM-PG and lower firmness of the corresponding hydrogels. Scanning electron microscope observed that the hydrogels exhibited a porous and interconnected morphology. The swelling ratio and volume of hydrogels was low at pH 3–5 and then became larger at pH 6–8 due to electrostatic repulsion resulting from deprotonated carboxymethyl groups. Particularly, the hydrogel prepared from chain-elongated CM-PG (DS = 0.25) showed the highest sensitivity to pH. These results suggested that phosphorylase-treated CM-PG formed the pH-responsive hydrogel and that the elongation degree and the properties of hydrogels depended on the carboxymethylation degree. Thus, it was inferred that these hydrogels was a potential carrier system of bioactive substances for their targeted releasing in small intestine.

## 1. Introduction

Hydrogel is composed of hydrophilic polymers that form the three-dimensional structure through chemical or physical cross-linking, which can absorb and retain a high percentage of water or biological liquids. Chemical cross-linking usually involves the chemical reaction of complementary groups on natural or synthetic polymers [1,2]. The defects of this technique include the harsh reaction conditions and the use of toxic reagents. In addition, the low biodegradability and biocompatibility of hydrogels made from synthetic polymers, such as poly (*N*-isopropylacrylamide) and poly (2-hydroxyethylmethacrylate), has limited their applications in food and biomedicine industry [3]. Physical hydrogels are formed due to secondary side inter- and intramolecular interaction, such as hydrogen bonds, electrostatic interactions or hydrophobic interactions, along with physical entanglements of the polymeric chains [4]. By contrast, hydrogels from physical cross-linking of natural polymers, such as starch and sodium alginate, are environmentally friendly. Moreover, intelligent hydrogels have arose the growing interest for applications in controlled release of active ingredients, which can undergo reversible volume change in response to small alterations of external environmental stimuli, such as pH, temperature, magnetic field, electric field, etc. [5]. Among these intelligent hydrogels, pH-responsive hydrogels are usually used to construct the controlled release system of bioactive ingredients in vivo based on the pH difference of gastrointestinal tract in the human body.

The pH-responsive hydrogels are usually constructed by polyelectrolytes containing dissociable groups [6]. The charge of these groups differs and thus the hydrogels absorb different amounts of water at different pHs. For instance, carboxymethyl starch and carboxymethyl chitosan have been used to construct the pH-responsive hydrogel [7,8,9]. However, the phytoglycogen-based hydrogel has been rarely reported. Phytoglycogen is a plant-based carbohydrate polymer that plays an important role in energy storage [10]. In phytoglycogen molecules, there are no long chains connecting individual clusters, an essential feature of amylopectin molecules [11]. Thus, phytoglycogen is a kind of amorphous hyperbranched glucan. It is widely known that the glucan chain with long enough length in water spontaneously self-assembles and forms the double helix through hydrogen bonds [12,13]. On the other hand, it was found that effective molecular entanglements were established in debranched starch owing to the strong hydrogen bonds between linear short amylose molecules, resulting in a stronger gel network and better resistance to amylase hydrolysis [14,15]. Thus, we speculated that once the chains of phytoglycogen were elongated, the elongated chains might form the double helixes that acted as cross-linking points for self-assembling hydrogel. Glycogen phosphorylase catalyzes the reversible phosphorolysis of α-(1, 4)-glucans at the non-reducing end, releasing a glucose-1-phosphate (Glc-1-P) [16]. Through the reversible reaction, α-(1, 4)-d-glucosidic linkages can be formed by the phosphorylase-catalyzed glycosylation in the presence of excess Glc-1-P [17,18]. To make this reaction occur, the primer with degree of polymerization (DP) ≥ 4 is required, and a glucose unit is transferred from Glc-1-P to a non-reducing 4-OH terminus of the primer [19]. Then the reactions take place successively, leading to formation of α-(1, 4)-glucan chains. On the other hand, it was found that carboxymethyl phytoglycogen had different zeta-potentials at different pHs [20], which suggested its sensitivity to pH for the potential application as the skeleton of the pH-responsive hydrogel. Therefore, this work aimed to develop pH-responsive hydrogels through combination of carboxymethylation and phosphorylase elongation of phytoglycogen (Figure 1) and study the structure and functional properties.

## 2. Materials and Methods

### 2.1. Materials

Phytoglycogen (PG) was extracted from sugary-1 maize kernels (Chinese Academy of Agricultural Sciences, Beijing, China). Glucose-1-phosphate (Glc-1-P) and glycogen phosphorylase were purchased from Sigma-Aldrich Chemical Co. (St. Louis, MO, USA). All other analytical-grade chemicals were obtained from Sinopharm Chemical Reagent Co. (Shanghai, China).

### 2.2. Preparation of pH-Responsive Hydrogels

Carboxymethyl phytoglycogen (CM-PG) with different degree of substitution (DS) at 0.15, 0.25, 0.30 and 0.40 were prepared as described in the previous study [20]. The PG was also considered as CM-PG with DS at 0 for convenience. The CM-PG with different DS at 0, 0.15, 0.25, 0.30 and 0.40 was dissolved in the phosphate buffer solution (0.01 mol/L, pH 7.0) to obtain the 8.0% (*w*/*v*) solution, and Glc-1-P was added into the solution with the ratio of CM-PG to Glc-1-P at 1:6. Glycogen phosphorylase (1.4 U/mL, 0.8 mL) was added and the volume of the final system was adjusted to be 4 mL using the phosphate buffer solution. The phosphorylase-catalyzed chain elongation occurred in a water bath at 40 °C for 24 h. Subsequently, the reaction mixtures were cooled to room temperature and allowed to stand at room temperature for 24 h to observe formation of hydrogels. The hydrogels were named Gel 1, Gel 2, Gel 3, Gel 4 and Gel 5, respectively. The resultant hydrogels were also dialyzed and lyophilized for further analysis.

### 2.3. Iodine Staining Analysis

The CM-PG or the lyophilized hydrogels (5 mg) were dispersed in 1 mL deionized water and 0.5 mL NaOH solution (1.0 mol/L) was added to dissolve the lyophilized hydrogels. The pH of the solution was adjusted to neutrality by HCl solution (1.0 mol/L) and the volume of the final system was adjusted to 5 mL with the deionized water. The sample solution (1 mL) was diluted by 5 times and 0.1 mL of the dilute iodine solution (0.1% iodine, 1% potassium iodide) was immediately added. After the mixtures stood for 15 min, the absorption spectra and the maximum absorption wavelength (λ_max_) were obtained by scanning in the range from 400 to 800 nm by a UV-Vis spectrophotometer.

### 2.4. X-ray Diffraction (XRD) Analysis

The crystal structure of samples was determined by an X-ray diffractometer (Bruker D2, AXS Co. Ltd., Karlsruhe, Germany). The XRD analysis was operated at 40 kV and 200 mA with Cu Kα radiation (λ = 0.1541 nm). The lyophilized gels and CM-PG powders were measured in a sample dish and scanned at a rate of 0.05°/s from 2θ 4° to 40° at the room temperature.

### 2.5. Scanning Electron Microscope (SEM) Analysis

The lyophilized hydrogels were fixed on the sample stage, and the cross-section was treated with a thin gold layer. The surface morphology of the lyophilized hydrogels was observed in a scanning electron microscope (Quanta-200, FEI Company, Eindhoven, The Netherlands) at the accelerating voltage of 5.0 kV.

### 2.6. Assay of Zeta-Potential

The zeta-potential measurement was conducted using a Zetasizer nano ZS (Malvern Instruments Ltd., Malvern, UK) at room temperature. The lyophilized hydrogels were completely dissolved in the NaOH solution (1.0 mol/L) and adjusted to different pHs (3–10) with the HCl solution (1.0 mol/L). The refractive index of the dispersion phase and particle used were set as 1.33 and 1.53, respectively.

### 2.7. Swelling Degree and pH-Responsive Behavior

The lyophilized hydrogels (50 mg) were dispersed in 1.0 mL deionized water and buffer solutions with different pHs. The obtained solutions were kept at room temperature for 24 h to form the hydrogel. The resultant hydrogels were centrifuged to remove the supernatant, and the residual moisture on the surface of swollen hydrogels was removed by the filter paper. The mass of the swollen hydrogel was accurately measured to calculate the swelling degree. The swelling degree was the mass ratio of swollen hydrogels to the lyophilized hydrogels. In addition, the lyophilized hydrogels (75 mg) were dispersed in 1.5 mL buffer solutions with different pHs and the hydrogels prepared at different pHs were photographed.

### 2.8. Rheological Properties

The rheological properties were determined using the Discovery HR-3 Rheometer (TA Instruments, New Castle, DE, USA) with plate geometry (40 mm diameter, gap height 1 mm). The temperature was kept at 25 °C using a circulating bath and a controlled peltier system. Frequency sweep test was performed, in which the frequency and strain were 0.1–100 Hz and 0.1%, respectively. The storage modulus (G′) and loss modulus (G″) were calculated from the strain response.

### 2.9. Mechanical Properties

The mechanical properties of hydrogels were measured by a texture analyzer (TA-XT2i, Stable Micro Systems Ltd., Godalming, UK) fitted with a cylindrical probe (P5). The elastic modulus program was used. The test conditions were as follows: the trigger force before the test at 5 g, the deformation at 40%, the speed before the test at 1 mm/s, the test speed at 1 mm/s, and the speed after the test at 10 mm/s.

### 2.10. Statistical Analysis

All data were performed in triplicate and the results were expressed as the mean values ± standard deviations. Data were analyzed using one-way analysis of variance (ANOVA) procedure using the Origin 8.0 (Origin Lab Inc., Northampton, MA, USA). A level of 0.05 was set to determine statistical significance.

## 3. Results and Discussion

### 3.1. Iodine Staining Analysis

According to the enzymatic polymerization mechanism of glycogen phosphorylase, the surface glucan chains of CM-PG would be elongated. Glucan with enough long length has the ability to complex with iodine, and the resultant complex is colored [21]. Moreover, the λ_max_ and the intensity of the absorption peak are positively correlated with the average chain length and the glucan content, respectively. Thus, iodine staining analysis was used to identify enzymatic chain elongation of CM-PG. The λ_max_ of the PG-iodine complex was 464.5 nm (Figure 1a). After PG was carboxymethylated, the λ_max_ did not significantly change, but the intensity of the absorption peak decreased as the DS increased, which suggested that carboxymethyl groups inhibited complexation between glucan and iodine. However, the λ_max_ of the enzymatically-modified CM-PG-iodine complex was 581.5 nm, 580.0 nm, 578.0 nm, 570.1 nm and 566.0 nm, respectively, when the DS of CM-PG was 0, 0.15, 0.25, 0.30 and 0.40, respectively (Figure 1b). These results suggested that treatment by phosphorylase increased the glucan length of CM-PG. Moreover, the λ_max_ of the enzymatically-modified CM-PG-iodine complex decreased gradually as the DS increased, which suggested that the average chain length of glucan decreased with the increasing of DS. In addition, the intensity decreased as the DS increased. Therefore, it was inferred that the higher DS resulted in less elongation of glucan chains. This was due to that the non-reducing ends of glucan might be substituted by carboxymethyl groups [20]. As a result, the carboxymethyl groups provided steric hinder for binding of glycogen phosphorylase on the surface of CM-PG and further elongation of the glucan chain.

### 3.2. XRD Analysis

No diffraction peak was observed in the X-ray diffraction pattern of CM-PG (Figure 2a), which confirmed that CM-PG was amorphous. After the phosphorylase-treatment of CM-PG, the hydrogels displayed strong diffraction peaks at 14.9°, 16.9°, 21.9° and 24.1° (Figure 2b), which were similar to the typical B-type crystal structure of starch [12,13]. The B-type crystal indicated that elongated glucan chains on the surface of CM-PG particles were interconnected and formed the double helix through hydrogen bonds. In summary, iodine staining and XRD analysis confirmed that glucan chains of the CM-PG molecules were successfully elongated through the action of glycogen phosphorylase, and elongated glucan chains formed the double helix. These results were in accordance with the previous study [22], in which the chain of glycogen was elongated by phosphorylase and then the elongated chains self-assembled and formed the double helix through hydrogen bonds. Moreover, enzymatic glucan grafting was successfully performed on chitosan [23,24], cellulose [25] and carboxymethyl cellulose [26].

### 3.3. Formation of the pH-Responsive Hydrogel

As stated above, the enzymatic chain elongation of CM-PG was catalyzed by phosphorylase in the phosphate buffer at 40 °C for 24 h. When cooled to room temperature and allowed to stand at that temperature, the resulting solutions turned into hydrogels. It was speculated that this gelation phenomenon was caused by gradual formation of junction zones, which were composed of double-helixes formed between the elongated glucan chains from the different CM-PG molecules. The chain-elongated PG formed the hydrogel at the fastest speed and the corresponding solution gelled once the ambient temperature decreased to room temperature (Appendix A). As the DS increased, the gelation rate of the corresponding solution slowed down. As stated above, the length of elongated chains on the surface of CM-PG and their content were negatively correlated with the DS of CM-PG. Thus, it was inferred that the gelation rate was positively correlated to the chain length of the elongated chains and their content. The longer chains and higher long chain contents were favorable for formation of the junction zones, thus accelerating gelation of chain-elongated CM-PG. Similarly, debranching starch could result in the stronger hydrogel through promoting formation of double helices stabilized by hydrogen bonds [14,15]. On the other hand, deprotonated carboxymethyl groups provided the repulsive force and steric hindrance for formation of double helix between glucan chains. Thus, higher DS of CM-PG slowed down gelation of chain-elongated CM-PG. As a result, the gelation rate was negatively correlated with the DS of chain-elongated CM-PG.

Lyophilziation of hydrogels readily resulted in xerogels with porous morphology (Figure 3). Gel 1 had a compact three-dimensional network structure. Compared with Gel 1, Gel 2–5 displayed relatively looser structure and larger pores. That is, as the DS increased, the porosity of the network structure in the gel was larger, and the cross-linking was looser. As stated above, carboxymethyl groups distributed on the surface of CM-PG inhibited enzymatic chain elongation. For this reason, fewer glucan chains on the CM-PG surface were elongated and the elongated glucan chains were much shorter. Thus, this result indicated that the hydrogels prepared from chain-elongated CM-PG with higher DS were composed of looser networks due to the smaller number of junction zones, which was also in accordance with the effect of DS of CM-PG on the gelation rate. On the other hand, the repulsive force between deprotonated carboxymethyl groups might also contribute to the looser structure of hydrogels.

### 3.4. Zeta-Potential of Hydrogels at Different pHs

The ζ-potential can be used to characterize the charge at the surface of polymers and it is a measure of the strength of mutual repulsion between particles. The zeta potential of all the samples was negative (Figure 4), indicating that all the samples had negative surface charges. Thus, lower ζ-potential indicated stronger repulsion herein. Carboxymethyl groups contributed to the surface negative charge of Gel 2–5. Therefore, Gel 1 had the highest ζ-potential and the pH had no significant influence on the ζ-potential of Gel 1 due to lack of carboxymethyl groups. However, the ζ-potential of Gel 2–5 significantly decreased when the pH changed from 3.0 to 7.0. This was due to that carboxymethyl groups were gradually deprotonated. Afterward, when the pH increased from 7.0 to 10.0, the ζ-potential did not significantly alter, which suggested that carboxymethyl groups were completely deprotonated at pH 7.0. At any pH, the ζ-potential of Gel 2–5 was negatively correlated with the DS due to more carboxymethyl groups. Accordingly, the ζ-potential of the hydrogel with the higher DS might be more sensitive to the change of pH. These results suggested that Gel 2–5 might be pH-responsive, while Gel 1 was not due to lack of pH-responsive carboxymethyl groups.

### 3.5. Swelling Degree and pH-Responsive Behavior

The swelling degree is an important parameter of hydrogels, which is influenced by both the cross-linking density and the intermolecular interaction between water and polymer segments [27]. Generally, larger swelling degree indicates a lower cross-linking density and higher polymer hydrophilicity. The swelling degree of Gel 1–5 was 7.29, 6.40, 4.87, 4.19 and 4.13 g/g, respectively (Figure 5a). As the DS increased, the swelling degree of hydrogels gradually decreased, although there were more hydrophilic carboxymethyl groups in hydrogels with higher DS. Therefore, it was inferred that the number of cross-linked regions played a dominant role in the water holding capacity of the hydrogel, which accorded with the previous study [28]. It was reported that the swelling degree of semi-interpenetrating polymer networks mainly depended on the amount of interpenetrating chains which acted as the cross-linking points. Herein, the glucan chains worked similarly to interpenetrating chains in semi-interpenetrating polymer networks. SEM analysis demonstrated that more cross-linked regions were produced form glucan chains on the surface of the CM-PG molecules with lower DS, which strengthened the pore density of the network, captured more water molecules and eventually resulted in higher swelling degree. The swelling degree of hydrogels at different pHs is demonstrated in Figure 5b. When the pH ranged 3–10, pH had no significant influence on the swelling degree of Gel 1, which was in accordance with the sensitivity of its ζ-potential to pH. In contrast, the swelling degree of Gel 2–5 first increased and then did not significantly change, which was consistent with the phase transition behavior of anionic gels [29]. Gel 2–5 contained carboxymethyl groups. Most of the carboxymethyl groups were protonated under low pHs, forming hydrogen bonds with hydroxyl groups of glucan chains. Thus, the polymers were not able to absorb a large amount of water, leading to low swelling degree. As the pH increased, more carboxymethyl groups deprotonated, resulting in stronger electrostatic repulsion. The electrostatic repulsion between the negative-charges broke hydrogen bonds and enhanced the hydration of the polymer [30], thus enlarging the network pore size of the hydrogel. As a result, the hydrogel was transformed into a loose state and absorbed more water, thus leading to high swelling degree. When the pH was 7.0, the deprotonating degree of carboxymethyl groups reached the maximum value and the swelling degree of hydrogels was the highest. Therefore, further increasing the pH did not significantly increase the swelling degree. These results further confirmed the pH-responsive behavior of Gel 2–5 and that the pH-responsive behavior resulted from the carboxymethyl groups. However, the sensitivity of Gel 2–5 disappeared when the pH was more than 7.0.

The pH-responsive behavior of hydrogels was also observable in Figure 6. The volume of Gel 1 was similar at different pHs. However, the volume of Gel 2–5 was pH-responsive. Specifically, Gel 2–5 was in a state of tight contraction at pH 3–5, while their swelling volume increased remarkably at pH 6–8, which was in accordance with the response of the swelling degree and zeta potential to pHs. These results suggested that Gel 2–5 might be the potential delivery system of bioactive ingredients for their targeted, releasing in the small intestine. However, the sensitivity of volume of hydrogels to pH was not positively correlated to the DS of hydrogels. It was obvious that the pH-responsive behavior of Gel 3 was the most significant. As mentioned above, the cross-linking density and swelling degree of hydrogels decreased as the DS of hydrogels increased, which might lead to the lower sensitivity of the volume of Gel 3–5 to pHs than Gel 2. On the other hand, although the cross-linking density and swelling capacity of Gel 2 were stronger than those of Gel 3, Gel 2 had fewer carboxymethyl groups than Gel 3, thus affecting its pH-responsive behavior.

### 3.6. Rheological Properties

In the frequency range of 0.01–100 Hz, the storage modulus (G′) of hydrogels was significantly higher than the loss modulus (G″) (Figure 7), suggesting that the elastic behavior dominated in hydrogels and the hydrogels were in the viscoelastic solid state [31]. As the frequency increased, the G′ and G″ of hydrogels were steady, indicating that the molecular chains in the gel system recovered the original three-dimensional network structure quickly after deformation and possessed strong resistance to the strain. Higher G′ indicated lower grid size and higher gel strength [32,33]. The G′ decreased as the DS increased. Thus, it was inferred that the grid size became smaller and the gel strength decreased as the DS increased, which were in accordance with the results of SEM analysis. The skeleton of hydrogels was made from interconnected glucan chains, and glucan chains were densely distributed in the hydrogel system prepared from CM-PG with lower DS, which restricted the movement of molecular chains to some extent. Hence, the corresponding hydrogel owned stronger resistance to shear strain and showed more obvious viscoelasticity of solid-like materials.

### 3.7. Mechanical Properties

The stress/strain curves of hydrogels under the compression mode are shown in Figure 8. When the strain was less than 10%, the stress increased slightly. Subsequently, the stress increased dramatically as the strain increased. The larger gradient of the curve meant that the corresponding hydrogel generated greater stress, which owned higher strength and was more brittle [29]. Accordingly, the firmness of Gel 1–5 was 875.6, 658.8, 545.3, 393.1 and 350.2 g, respectively. That is, the hydrogel firmness was negatively correlated with the DS (*p* < 0.05), which was in accordance with the result of rheology analysis and the order of the gelation rate. This was probably due to formation of looser networks. As shown in the SEM images, fewer junction zones were formed in hydrogels with higher DS. On the other hand, the repulsion force resulting from negative-charged carboxymethyl groups might also decrease the gel firmness. In summary, the DS of CM-PG significantly affected the glucan chain length and the glucan content of chain-elongated CM-PG, thus evidently influencing the gelation rate of chain-elongated CM-PG and the properties of the corresponding hydrogel.

## 4. Conclusions

Phytoglycogen was carboxymethylated and chain-elongated by glycogen phosphorylase. Consequently, the elongated glucan chains from different carboxymethyl phytoglycogen (CM-PG) molecules formed double helices, which acted as cross-linking points to produce the hydrogel. As the DS of CM-PG increased, the content of elongated glucan reduced, and the average elongated chain length shortened. Thus, the gelation rate of the corresponding chain-elongated CM-PG was slower. Due to the presence of carboxymethyl groups, the hydrogels exhibited the pH-responsive behavior. Specifically, the hydrogel volume was small at pH 3–5 and became larger at 6–8, which suggested that these hydrogels was a potential carrier system of bioactive ingredients for their targeted releasing in small intestine. Therefore, the application of pH-responsive hydrogels based on chain-elongated CM-PG in encapsulating bioactive ingredients will be investigated in future.

## Data Availability

The data presented in this study are available in the article and its Appendix A.

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
