# Peer review of "Fabrication, Structure and Functional Characterizations of pH-Responsive Hydrogels Derived from Phytoglycogen"

_foods, 2021, doi:10.3390/foods10112653_

Round 1

Reviewer 1 Report

This manuscript describes the elaboration and the characteristics of hydrogels from carboxymethyl-phytoglycogen after elongation by the phosphorylase-catalyzed glycosylation in the presence of excess glucose-1-phosphate.

The results are well described but could be improved or completed (see below). I suggest major revisions.

In the introduction, the authors indicate that chemical cross-linking involves polymerization of monomers (lines 31-32) but many chemical cross-linking reactions can be achieved without polymerization reactions, in particular in the polysaccharide’s chemistry. Many examples can be found: for example, the cross-linking of polysaccharides with sodium trimetaphosphate (ref: Polymer Bulletin 2004, 52, 429-436, DOI 10.1007/s00289-004-0299-4), the cross-linking with acid adipic dihydrazyde (Journal of Controlled Release, 2000, 69, 169-184, DOI 10.1016/S0168-3659(00)00300-X) etc… So, this part should be modified. The authors should explain what the phosphorylation-catalyzed reaction is in the introduction because without this reaction the formation of the gel does not occur.

The preparation of hydrogels is singular. A scheme explaining the elongation of the chains by the phosphorylation-catalyzed reaction and the formation of the hydrogels will help the understanding for anyone unfamiliar with this reaction.

The effect of carboxymethylation is first studied by iodine staining to show the influence of DS on elongation. The authors bring the conclusion of the effect of DS on elongation from the first lines of part 3.1 (lines 153-154), which questions on how it works? They bring the explanations in the following lines. This disrupts the reading and understanding of this part.

The characterization of the phosphorylase-elongated CM-PG before the formation of the hydrogels is only described by iodine staining analysis or XRD analysis. Did the authors try to measure their molecular weights or their rheological properties?

In part 3.3 (line 198), sodium acetate buffer is mentioned for the formation of the hydrogels while phosphate buffer is mentioned in materials and methods (part2.2 line 86). Which buffer is the good one?

The scales of Fig.3 are not clear and visible. Gel 5 is not depicted on Fig.3 (I and J are missing)

Part 2.6. The zeta-potential of the hydrogels is measured. The authors mentioned that hydrogels are completely dissolved in solutions with different pH: are the hydrogels really dissolved losing their 3D structure or are they swelled? What kind of solutions are used? In part 3.4, the authors explain the negative surface of Gel 1 (without negative carboxylate groups) by the presence of hydroxyl groups. Are the hydroxyl groups under their alkoxide form? Why?

Part 3.6. Line 316: the frequency range is 0.01-100 Hz (as seen on Fig. 7) instead of 0.1-100 Hz; Line 319: why the authors say that G’ and G’’ are steady when the frequency increases while G’ and G’’ continually increase with frequency Lines 323-324: the sentence “Thus, it was inferred….as the DS increased” is not correct: if DS increases, G’ decreases indicating a decrease of the hydrogel strength. The authors should use the same scale for G’ and G’’ on Fig.7, if not, the interpretation of the curves can be wrong.

Reviewer 2 Report

The Authors presented studies of fabrication, structure, and functional characterizations of phytoglycogen-derived pH-responsive hydrogels. The topic is "hot", but in this study, a lot of information is missing and there is a need to add an additional evaluation. For that reason, the manuscript is not recommended for publishing at this stage. What is more, the connection with "food" is weak, and this point should be also enhanced. Detailed comments are listed below: 

Abstract

Add information about the characterization techniques which were applied for this study

In the final part, please add some perspectives of your results for further applications/studies

Introduction

most of the introduction is written too general. Please add some examples (names) of the structures that you discussed here. For me, a lot of the presented data is suitable for PNIPAM, which is synthetic, so please give examples. Especially, the examples which correspond with your study's dedicated "natural base" materials.

Materials and methods

  • just a brief description of the samples preparation is needed. Please add a table where detailed samples description will be presented (it will improve the quality of the paper for readers)
  • p. 2.6 add details about the solutions. 
  • p. 2.7. what was the total volume of the samples?
  • Please consider adding additional test results such as FTIR - it will be more suitable than UV-Vis 

Results and discussion 

  • only XRD studies for me did not fully prove predicted structural changes and additional chemical tests should be provided. Nevertheless, some parts like SEM images, mechanical and rheological tests are well described. Unfortunately, the results of swelling activity, as well as pH sensivity, were not fully proven. Additional tests, comments, and comparisons with other studies are necessary. 

Round 2

Reviewer 1 Report

No further comments

Reviewer 2 Report

The Authors mostly improved the manuscript content due to the Reviewers comments. Additional Authors' comments clarified an idea of the manuscript. Still, minor improvements are necessary, but overall I m recommending this manuscript for publishing after the listed below changes.

l. 18 "Scanning electron microscope observed" --> SEM showed

An additional table with gel samples (gel 1, 2...) is necessary (i.e. in the materials and methods section or at the beginning of the result and discussion section).

Finally, please summarize the most important achievements i.e. which sample was optimal in the conclusions.